# Horn-like space-coiling metamaterials toward simultaneous phase and amplitude modulation

Reza Ghaffarivardavagh [1], Jacob Nikolajczyk [1], R. Glynn Holt [1], Stephan Anderson[2] & Xin Zhang [1]

Acoustic metasurfaces represent a family of planar wavefront-shaping devices garnering increasing attention due to their capacity for novel acoustic wave manipulation. By precisely tailoring the geometry of these engineered surfaces, the effective refractive index may be modulated and, consequently, acoustic phase delays tuned. Despite the successful demonstration of phase engineering using metasurfaces, amplitude modulation remains overlooked. Herein, we present a class of metasurfaces featuring a horn-like space-coiling structure, enabling acoustic control with simultaneous phase and amplitude modulation. The functionality of this class of metasurfaces, featuring a gradient in channel spacing, has been investigated theoretically and numerically and an equivalent model simplifying the structural behavior is presented. A metasurface featuring this geometry has been designed and its functionality in modifying acoustic radiation patterns experimentally validated. This class of acoustic metasurface provides an efficient design methodology enabling complete acoustic wave manipulation, which may find utility in applications including biomedical imaging, acoustic communication, and non-destructive testing.

[1] Department of Mechanical Engineering, Boston University, Boston, MA 02215, USA. [2] Department of Radiology, Boston University Medical Campus, Boston, MA 02118, USA. Correspondence and requests for materials should be addressed to S.A. (email: sande@bu.edu) or to X.Z. (email: xinz@bu.edu)

Acoustic wavefront modulation is of great interest given the numerous promising applications such as acoustic communication and biomedical imaging, among others. Traditionally, acoustic wavefront modulation has been realized in the context of phased array transducers, which, due to their complexity, are considered expensive in terms of both design and implementation aspects. In recent years, following successful achievements in electromagnetic wave manipulation using metamaterials and metasurface science[1–4], ongoing efforts have been made to utilize metasurfaces for acoustic wavefront modulation[5–11]. Metasurfaces, engineered surfaces of subwavelength thickness, offer a unique approach to acoustic manipulation in which a desired wave pattern may be achieved by precisely designing the constituent unit cell structures. Among the unit cell structures reported to date, space-coiling structures[9, 10, 12–17] are drawing growing attention due to their incredibly simple structure, ease of fabrication, and demonstration of successful wavefront manipulation capacity. In these structures, the space-coiling geometry is designed to generate the desired phase shift in the radiated acoustic signal, while mitigating the impedance mismatch in order to optimize power transmission and amplitude uniformity[18]. Despite the fact that low transmission loss in such phase-based wavefront modulation yields high conversion efficiency[19], the complexity of the design process required to obtain a desired metasurface phase map remains a fundamental limitation. Typically, iterative or stochastic algorithms are required to derive the ideal phase map, relying on an appropriate initial guess in order to reach the global minimum[20]. Moreover, transitioning from two-dimensional (2D) to three-dimensional (3D) phase-based wavefront manipulation approaches radically increases the design cost and complexity.

In the context of complete acoustic wave modulation, in which both phase and amplitude are simultaneously modulated, the additional degree of freedom leads to a marked simplification of the metasurface design process. As opposed to complicated optimization procedures required with phase-based wavefront modulation, complete acoustic wave modulation may leverage the time-reversal technique in which the required phase and amplitude map may be readily obtained. Furthermore, by extending the properly designed unit cell structure to the 3D domain without additional complexity, acoustic metamaterial bricks or holograms may be generated to render 3D acoustic field patterns.

Although prior work has focused primarily on phase engineering, herein we demonstrate the possibility of simultaneous phase and amplitude modulation in a class of horn-like space-coiling metasurfaces. Utilizing a gradient in channel spacing of the unit cells, full acoustic control over wavefront modulation is realized. We first demonstrate that conventional space-coiling metamaterials possess a fundamental limitation in the

simultaneous modulation of both phase and amplitude of transmitted wave acoustic waves. Specifically, the complex transmission through conventional space-coiling structures is shown to be inherently bounded in certain complex regions. Subsequently, we extend the applicability of the presented bound to the general case of the metasurface and infer the necessary conditions for the realization of complete acoustic wavefront modulation. Ultimately, a modified, horn-like space-coiling metamaterial structure is presented that satisfies the necessary conditions for complete acoustic wave modulation. Finally, the functionality of the horn-like space-coiling metamaterial structure is theoretically and experimentally validated with regards to its capacity for simultaneous phase and amplitude modulation. This study seeks to shift the paradigm in acoustic metasurfaces through the realization of simultaneous control of phase and amplitude, thereby paving the way for a new generation of acoustic devices.

## Results

**Complete wave modulation in space-coiling metamaterials.** The concept of space-coiling metamaterials was initially proposed by Liang et al.[21] It was demonstrated that acoustic waves with frequencies above a given cutoff value would propagate along an elongated path within an assembly of zigzag channels (Fig. 1a). The elongated path of the acoustic wave leads to the occurrence of a phase delay in the transmitted wave and, consequently, a higher refractive index is realized. Moreover, phenomena such as negative refractive index[22], zero index[23], and Dirac-like dispersion[24] have been also demonstrated using space-coiling structures. Space-coiling metamaterials possess marked advantages due to their simple structure and ease of design. It has been demonstrated that when the channel width ($d$) is sufficiently small with respect to the wavelength, the relative refractive index of the coiled structure ($n_r$: effective refractive index of coiled structure normalized by the original fluid index) can be precisely calculated using the path length of the acoustic wave shown in Fig. 1a[21]. The relative refractive index can be expressed as:

$$n_r = \frac{L_{eff}}{t} \tag{1}$$

where $t$ is the overall length of the coiled structure and $L_{eff}$ can be estimated as:

$$L_{eff} \approx N \times L \tag{2}$$

where $N$ denotes the number of coils (for example, $N = 7$ in the structure depicted in Fig. 1a) and $L$ is the length of each branch and is approximated as:

$$L = \sqrt{(a - d)^2 + (d + w)^2} \tag{3}$$

Given the expression for the relative refractive index of the coil structure, it can be represented as an equivalent model of the same dimensions but comprised a straight channel filled with a medium of refractive index of $n_r n_0$ (shown in Fig. 1b) in which $n_0$ represents the refractive index of the original fluid. Please note that from another perspective, the space-coiling structure can also be modeled as a single straight channel structure similar to Fig. 1b and filled with the original fluid with refractive index of $n_0$ but with an overall length of $L_{eff}$. By employing the equivalent model of the space-coiling structure, one can derive the transmission coefficient, denoted as $T$, for a normally incident plane wave as follows:

$$T = |T|e^{i\theta} = \frac{4}{\left(1 + \frac{a}{d}\right)\left(1 + \frac{d}{a}\right)e^{-ik_0 n_r t} + \left(1 - \frac{a}{d}\right)\left(1 - \frac{d}{a}\right)e^{ik_0 n_r t}} \tag{4}$$

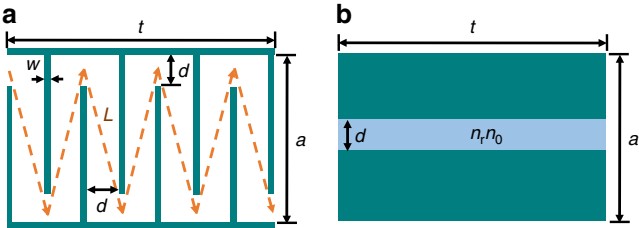

**Fig. 1** Traditional space-coiling metamaterial unit cell and its equivalent model. **a** Space-coiling metamaterial structure with the overall length of $t$ and overall width of $a$ is depicted here. $d$ and $w$ are channel width and coil's wall thickness respectively; acoustic wave trajectory is shown as the dashed line in which $L$ represents the wave trajectory length within each coil. **b** Equivalent model of the coil structure featuring a single straight channel filled with medium of different refractive index ($n_r n_0$)

where $k_0$ is wave number associated with the medium inside the zigzag channel, $a$, $d$, and $t$ (shown in Fig. 1a) represent the unit cell width, channel width, and unit cell length, respectively, and $n_r$ is the relative refractive index discussed above. Through the rearrangement of terms in Eq. (4) (see Supplementary Note 1), the relationship between the phase and amplitude of the transmission coefficient is derived as follows:

$$|T| = \sqrt{\frac{1 + \frac{\tan^2(\theta)}{S^2}}{1 + \tan^2(\theta)}} \quad (5)$$

$$S = \frac{1}{2}\left(\frac{a}{d} + \frac{d}{a}\right) \quad (6)$$

Considering the fact that the term $S$ in Eq. (6) is always greater than unity ($S \geq 1$), the amplitude of the transmission coefficient for any transmitted phase ($\theta$) is bounded by:

$$1 \geq |T| \geq |\cos(\theta)| \quad (7)$$

The lower bound occurs when the channel width ($d$) is significantly smaller than unit cell width ($a$). To further validate the presence of the aforementioned bound on sound transmission through a space-coiled structure, the transmission coefficient has been derived analytically using the transfer matrix method (TMM). In this method, by employing modal analysis and applying boundary conditions within each coil, the transfer matrix has been obtained and, consequently, the transmission coefficient of the entire unit cell has been derived. Additional information regarding the details of the TMM solution are discussed in Supplementary Note 2. Using the TMM solution and by varying the geometries of the traditional space-coiling structure, a large set of solutions has been investigated and the resultant transmission phases and amplitudes are depicted in Fig. 2. By comparing the bound (pink region) predicted by Eq. (7) resulting from the equivalent model and the set of results from the TMM-based solution, it can be observed that the set of results using the TMM fall within the predicted transmission bound, thereby ensuring the validity of the bound represented in Eq. (7). Given the presented transmission bound shown in Fig. 2, it is apparent that the conventional space-coiling structure cannot be used for simultaneous phase and amplitude modulation. The bounded nature of the transmission phase and amplitude results in large portions of the phase-amplitude diagram being inaccessible and

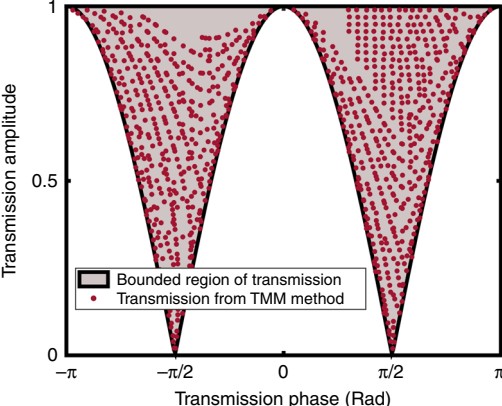

**Fig. 2** Phase and amplitude of the transmitted wave with conventional space-coiling structure. The colored region is the predicted space using the equivalent model. Scatter results shown by red dots are derived from the TMM-based approach by varying the geometry of the unit cell

simultaneous modulation of phase and amplitude is essentially precluded. Of note, however, the conventional space-coiling structure is capable of modulating the amplitude of the transmitted acoustic wave in specific instances. By tailoring the unit cell design and targeting a transmission phase of either $\pi/2$ or $-\pi/2$, the complete range of amplitude modulation is accessible (0–1). Nevertheless, the capacity for simultaneous and independent modulation of phase and amplitude is highly limited in the conventional class of space-coiling structures.

Importantly, the transmission bound presented herein is generalizable for any arbitrary metasurface and is not limited to space-coiling structures. Herein we investigate the origin of this transmission bound and analyze approaches to overcome this bound in order to yield full wavefront control below.

Considering a general metasurface with thickness of $L_m$, equivalent acoustic impedance of $Z_m$ and wave number of $K_m$ placed in infinite space with acoustic impedance of $Z$, one may derive the following:

$$1 - R = T\left[\cos(K_m L_m) + i\frac{Z}{Z_m}\sin(K_m L_m)\right] \quad (8)$$

$$1 + R = T\left[\cos(K_m L_m) + i\frac{Z_m}{Z}\sin(K_m L_m)\right] \quad (9)$$

In which $R$ and $T$ denote the reflection and transmission coefficients of the metasurface in a 2D space, respectively. Assuming a metasurface with purely real acoustic impedance ($Z_m$), the validity of the complex conjugate form of Eq. (9) is ensured.

$$1 + R^* = T^*\left[\cos(K_m L_m) - i\frac{Z_m}{Z}\sin(K_m L_m)\right] \quad (10)$$

in which the asterisks denote the complex conjugate operator. By multiplying the two sides of Eq. (8) and Eq. (10) with each other and assuming a passive, lossless metasurface ($|R|^2 + |T|^2 = 1$), the following may be derived:

$$R - R^* = i|T|^2\sin(K_m L_m)\cos(K_m L_m)\left(\frac{Z_m}{Z} - \frac{Z}{Z_m}\right) \quad (11)$$

In addition, by subtracting Eq. (8) from Eq. (9), it can be concluded that:

$$2R = iT\left[\sin(K_m L_m)\left(\frac{Z_m}{Z} - \frac{Z}{Z_m}\right)\right] \quad (12)$$

By substituting Eq. (12) into the Eq. (11), one may readily derive the following:

$$|T| = \frac{\cos(\theta)}{\cos(K_m L_m)} \quad (13)$$

in which ($\theta$) is the transmitted phase. Equation (13) demonstrates the general coupling between the transmission phase and amplitude from which the aforementioned bound shown in Eq. (7) can be inferred. Notably, this bound exists for any arbitrary, passive, lossless metasurface with real acoustic impedance, regardless of the internal structure. Please note that for ultrathin metasurfaces, $\cos(K_m L_m)$ goes to unity and the transmission amplitude will approach $\cos(\theta)$, similar to its electromagnetic counterpart's reported bound[25]. The results obtained herein indicate that the key element for realizing full wavefront modulation is the presence of the acoustic reactance term in the metasurface acoustic impedance. For metasurfaces with complex acoustic impedance, Eq. (10) is no longer valid and may be altered to an alternate form from which a similar bound

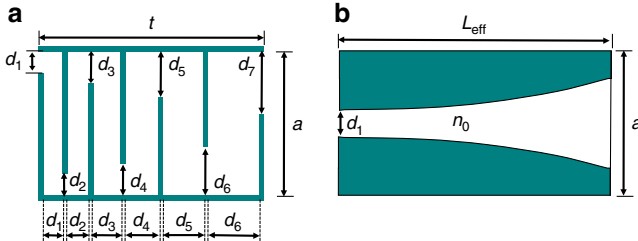

**Fig. 3** Horn-like space-coiling metamaterial unit cell and its equivalent model. **a** Horn-like space-coiling metamaterial structure composed of six coils is shown here, where $d_n$ ($n = 1{:}6$) represents the channel width at $n^{th}$ coil and $d_7$ is the channel width at the output port; channel widths follows the geometrical sequence with common ratio in excess of one (CR > 1). **b** Equivalent model of the gradient coil structure featuring a horn-shaped channel of length $L_{eff}$ filled with original medium with refractive index of ($n_0$)

cannot be derived. Hence, in order to realize full acoustic control of the transmitted wave, we present a horn-like class of space-coiling structures featuring a gradient in channel spacing that possess complex acoustic impedance and investigate its capacity for simultaneous phase and amplitude modulation.

**Horn-like space-coiling metamaterials**. To date, space-coiling metamaterial structures have been designed such that the zigzag channel width ($d$) remains constant throughout the length of the unit cell. In the previous section, we have demonstrated that this design methodology imposes a limited internal bound on acoustic transmission phase and amplitude, regardless of unit cell size or working frequency. Herein, a horn-like space-coiling structure is presented in which the gradual change in channel width leads to the presence of an imaginary term in the acoustic impedance of the metasurface. Although the variation in channel width may be realized in many distinct forms (linear, periodic, arithmetic, geometric, etc.), this study will focus on a unique case of geometric progression of channel width, which resembles a horn-like shape. The horn-like structure presented herein (shown in Fig. 3a), instead of having a constant channel width throughout its length, features a change in width at each step governed by a constant common ratio (CR), defined as:

$$CR = \frac{d_n}{d_{n-1}} \quad (14)$$

In which $d_n$ and $d_{n-1}$ are the channel width of $n^{th}$ and $n-1^{th}$ coil, respectively. By this definition, a structure with CR = 1 simply represents the conventional space-coiling metamaterial structure depicted in Fig. 1a. However, for structures with CR > 1, the zigzag channel can be well approximated as an exponential horn with a flare constant of:

$$m = \frac{N}{L_{eff}} \ln(CR) \quad (15)$$

where $N$ is the number of coils and $L_{eff}$ is the effective length of the zigzag structure, which can be calculated using the aforementioned acoustic path length (see Supplementary Note 3 for details of derivation). By employing the equivalent horn-like model (shown in Fig. 3b) to investigate the behavior of the gradient space-coiling structures, the acoustic transmission coefficient may be analytically derived using Webster's horn equation for velocity potential[26]:

$$\left(\frac{\partial^2}{\partial x^2} + m\frac{\partial}{\partial x} + k^2\right)\emptyset = 0 \quad (16)$$

where $\emptyset$ is the velocity potential, $m$ is the flare constant, and $k$ is wave number. The solutions of Eq. (16) are in the form of:

$$\emptyset = C_1 e^{\mu_1 x} + C_2 e^{\mu_2 x} \quad (17)$$

$$\mu_1 = -\frac{m}{2} + \frac{i}{2}\sqrt{4k^2 - m^2} \quad (18)$$

$$\mu_2 = -\frac{m}{2} - \frac{i}{2}\sqrt{4k^2 - m^2} \quad (19)$$

By calculating the pressure and velocity from the velocity potential and applying the boundary conditions at the horn's mouth and throat, the resultant transmission coefficient appears as (see Supplementary Note 3):

$$T = |T|e^{i\theta} = \frac{4}{\left(-\frac{k}{\beta}\frac{a}{d_{out}} + 1 + \frac{d_{in}}{d_{out}} - \frac{k}{\beta}\frac{d_{in}}{a} + i\frac{\gamma}{\beta}\left(1 - \frac{d_{in}}{d_{out}}\right)\right)e^{(\gamma + i\beta)L_{eff}} + \left(\frac{k}{\beta}\frac{a}{d_{out}} + 1 + \frac{d_{in}}{d_{out}} + \frac{k}{\beta}\frac{d_{in}}{a} - i\frac{\gamma}{\beta}\left(1 - \frac{d_{in}}{d_{out}}\right)\right)e^{(\gamma - i\beta)L_{eff}}} \quad (20)$$

where $d_{in}$ and $d_{out}$ are the channel width at the input and output ports, respectively, $a$ is unit cell width, $k$ is wave number, and $\gamma$ and $\beta$ are defined as:

$$\gamma = \frac{m}{2} \quad (21)$$

$$\beta = \frac{1}{2}\sqrt{4k^2 - m^2} \quad (22)$$

Eq. (20) represents the generalized form of the transmission coefficient for the gradient space-coiling structure with a geometric progression. It can readily be demonstrated that in the case of the conventional space-coiling structure in which the channel width is constant (CR = 1), the flare constant ($m$) will reach zero and Eq. (20) can be simplified to the form of Eq. (4). From Eq. (20), an expression relating phase and amplitude of the transmission coefficient, similar to Eq. (5), may be derived for the horn-like space-coiling structure as follows (see Supplementary Note 3):

$$|T| = \frac{2}{CR^{N/2} + CR^{-N/2}}\sqrt{\frac{1 + \frac{\tan^2(\theta)}{S^2}}{1 + \tan^2(\theta)}} \quad (23)$$

$$S = \frac{\left(\frac{a}{d_{out}} + \frac{d_{in}}{a}\right)}{\left(1 + \frac{d_{in}}{d_{out}}\right)} \quad (24)$$

It can readily be demonstrated that $S \geq 1$ and $CR^{N/2} + CR^{-N/2} \geq 2$; therefore, the transmission amplitude will be bounded:

$$\frac{2}{CR^{N/2} + CR^{-N/2}} \geq |T| \geq \frac{2}{CR^{N/2} + CR^{-N/2}}|\cos(\theta)| \quad (25)$$

Based on Eq. (25), both the upper and lower bounds of the transmission amplitude are functions of the CR and the number of coils ($N$), and increasing these two parameters results in lowering both bounds. Thus, in order to validate Eq. (25) and the bounds on the transmission coefficient for the horn-like space-coiling metamaterials, a limited range of the number of coils ($N = 1$–$15$) for four different values of the CR (CR = 1.1–1.4) are considered. In addition, from the TMM-based approach, the set of results for different unit cell geometries, in accordance with the aforementioned range of $N$ for each value of CR, have been obtained and are shown in Fig. 4. For each structure with a distinct CR, the upper bound is constructed with a value of $N = 1$, with a tiny decrease in the upper bound as a function of CR (e.g., $\approx 0.98$ for CR = 1.4). However, the lower bound is related to the

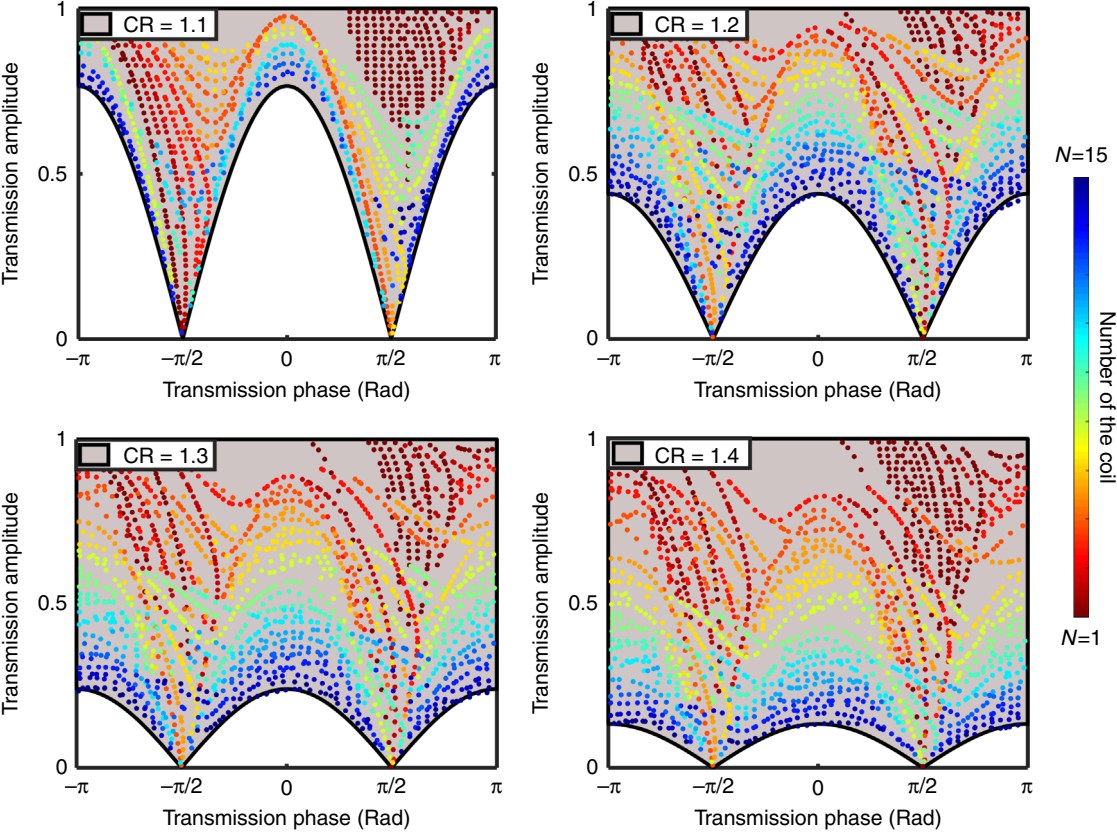

**Fig. 4** Illustration of the result space for gradient space-coiling metamaterials with varying common ratio (CR). The colored region (pink) is the region defined by Eq. (25) and the dot points represent the results from TMM

highest $N$ ($N = 15$, in this example analysis) and the effect of variations in CR is enhanced and the lower bound may be significantly decreased (e.g., $\approx 0.16$ for CR = 1.4). Of note, there exists a tradeoff between CR and the maximum allowable range of $N$ given the fact that channel width at the output port ($d_{out}$) must remain smaller than the unit cell width ($a$). As illustrated in Fig. 4, this class of horn-like space-coiling structures enables marked expansion of the coverage of the phase-amplitude diagram well beyond the inherent limits encountered in conventional space-coiling metamaterials. The expansion of the coverage of the phase-amplitude diagram yields the realization of full wavefront manipulation through simultaneous phase and amplitude modulation in gradient space-coiling metamaterials. Please note that expanding coverage of the phase-amplitude diagram is not the only criteria requisite to realizing full wavefront modulation using the proposed structure. Of critical importance is also the fact that, due to the classical and simple shape of the horn-like space-coiling metamaterials, the presentation of an analytical formulation is enabled, thereby easing the complexity of the design process.

**Metasurfaces for complete acoustic wave modulation.** Subsequent to demonstrating the capability of gradient space-coiling metamaterials to expand the accessible transmission phase-amplitude region, metasurfaces comprising these structures have been designed, targeting two distinct functionalities: sound focusing and acoustic beam splitting. The metasurfaces thusly designed herein feature 30 horn-like space coiling unit cells of identical width ($a = \lambda/6$) and length ($t = \lambda/2$) but distinct internal structures. Although constant width and length of the unit cells is not requisite to achieving the intended functionalities, these

constraints have been implemented to further demonstrate the ease of design using this class of horn-like space-coiling metamaterials. Please note that the unit cell's width has a critical role in conversion efficiency, with smaller widths being preferable for optimal performance (see Supplementary Note 4 for detailed discussion). The internal structure of each unit cell ($N$, $w$, $d_{in}$, and CR) has been designed to generate the desired phase and amplitude in order to shape the transmitted acoustic wavefront. The first step undertaken herein in designing the metasurfaces was to derive the required phase and amplitude of the transmitted wave from each unit cell of a particular metasurface. To this end, the concept of time-reversal or the phase-conjugation method in the frequency domain have been utilized, given the capability for simultaneous phase and amplitude modulation. Implementing the aforementioned approaches in designing the metasurface allows for a drastic reduction in the computational expense when compared with phase-based metasurfaces in which rigorous optimization is required. The desired transmission amplitude-phase profile of the metasurface with regards to a given functionality may be obtained from both numerical and analytical approaches. Herein, numerical techniques have been utilized for this step, as this approach may readily be generalized for any complicated profile, such as 3D acoustic holograms. Following the derivation of the requisite phase and amplitude of transmission at each unit cell, the internal geometry of the metasurface's unit cell has been designed analytically using the TMM approach and, ultimately, the entire metasurface in both lossy and lossless conditions has been simulated to visualize the targeted performance.

In the case of focusing of the acoustic wave shown in Fig. 5a, the metasurface has been designed to focus the sound at a focal point located two wavelengths from the metasurface. In order to

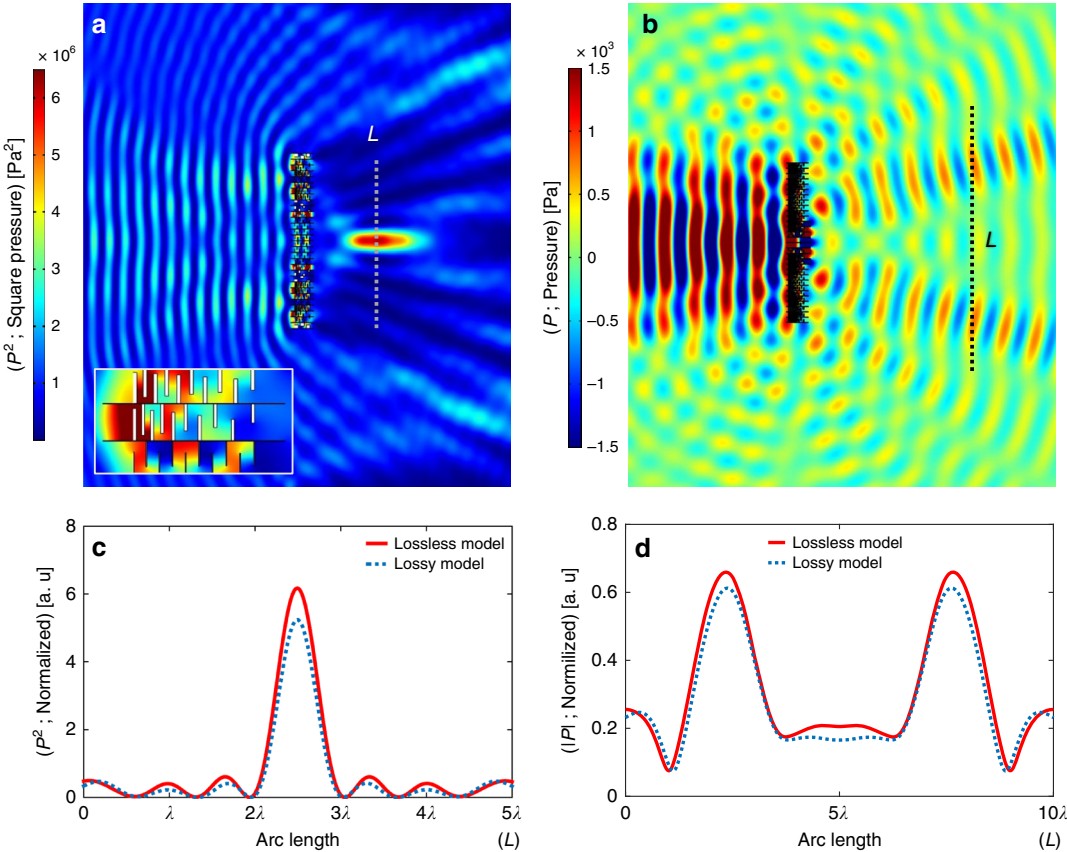

**Fig. 5** Horn-like space-coiling metasurface-based wavefront manipulation. **a** Acoustic plane wave with an amplitude of 1 kPa incident on the left-hand side of the metasurface and the resultant focusing on the right-hand side. **b** Acoustic beam splitter that transform the plane incident wave with the amplitude of 1 kPa on left-hand side of the metasurface to two equi-amplitude beams in $\pi/12$ and $-\pi/12$ directions. **c** Pressure profile along the cut-line with the length of $5\lambda$ extending through the focal spot shown in Fig. (5a). **d** Pressure profile along the cut-line placed at a distance of $7\lambda$ from the metasurface on split beam side shown in Fig. (5b). In Fig. 5c, d, the red line represents the data from the lossless simulation, while the blue dotted line represents the data with the presence of loss

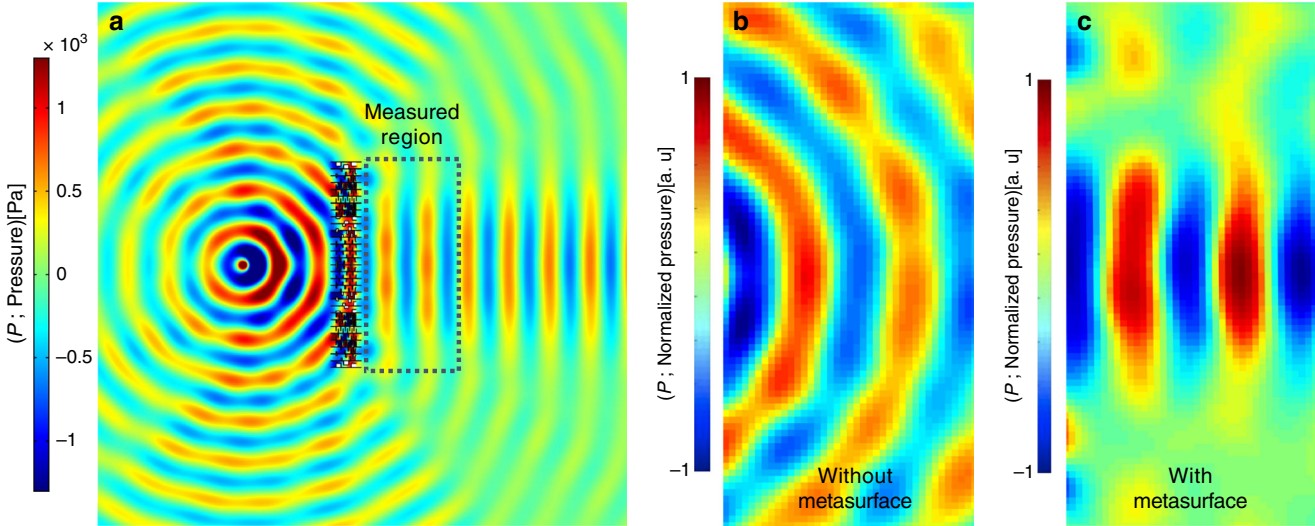

**Fig. 6** Numerical and experimental results of the reverse form of focusing. A monopole point source is placed two wavelengths away from the metasurface and a laterally confined plane wave is created on the transmission side. **a** Simulation results showing the cylindrical to plane wave conversion. Rectangular dotted region depicts location in which the experimental acoustic wave pattern was mapped. **b** Measured normalized acoustic field in the absence of the metasurface. **c** Measured normalized acoustic field with in the presence of the metasurface

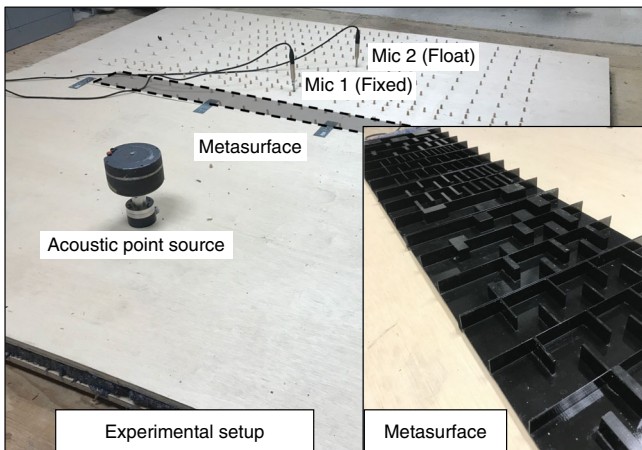

**Fig. 7** Experimental setup. Experimental setup employed to map the resultant acoustic field in the specified region on the transmission side of the metasurface

determine the proper wavefront, a hypothetical monopole point source has been considered on the desired focusing spot and the resultant complex conjugate pressure has been calculated over the transmission side of the metasurface (right-hand-side). Next, given the plane wave incident on the left-hand-side of the metasurface, the desired transmission coefficient for each unit cell is computed and the unit cell structures have been designed accordingly. With regards to the acoustic beam splitting shown in Fig. 5b, two plane waves with angles of $\pi/12$ and $-\pi/12$ have been assumed on the transmission side of the metasurface and the desired transmission coefficients at each unit cell have been calculated. The details of the unit cell geometry, along with the resultant transmission coefficient have been detailed in Supplementary Table 1.

The sound focusing results are shown in Fig. 5a in which the transmitted wave has been focused at the desired focal spot. As a common aim in focusing applications is the confinement of the acoustic wave power, the squared pressure ($p^2$) in Fig. 5a, c is depicted and corresponds to the resultant power. By employing the horn-like space-coiling design, a high degree of focusing has been obtained and the resultant focal region demonstrates a distinguishable high power region on the transmission side. In order to quantify the resultant focusing, $p^2$ is probed along the cut-line (shown in Fig. 5a) and is depicted in Fig. 5c. Notably, the pressure at the focal point is ~ 2.5 times the incident pressure, yielding a power confinement on the order of 6. Moreover, the pressure profile depicted in Fig. 5c successfully mimics the form of the Hankel function as the monopole source and a pressure power ratio of ~ 8 (6.3 peak at focal region and 0.8 at nearest peak to focal region) have been obtained. Finally, the acoustic beam splitting results are shown in Fig. 5b in which the normally incident plane wave on the metasurface has been divided into the two equi-amplitde beams in the desired directions. In order to quantify the results, an absolute pressure profile normalized by the incident beam's amplitude has been probed along the cutline and depicted in Fig. 5d. It can be observed that the split beams maintain an amplitude of ~ 65% of the incident beam and an amplitude resolution of ~ 3 (0.65 in the beam regions and 0.2 in the region between the 2 beams) may be achieved with the designed metasurface.

In order to experimentally validate the proposed design methodology, a metasurface featuring the horn-like space coiling metamaterial design has been fabricated and tested. To this end, for the sake of simplicity, the reverse form of the sound focusing case has been experimentally validated in order to avoid the

complexity associated with the practical generation of a plane wave in the near-field regime. Given the time-reversible nature of the proposed structure, if the point source is placed at a distance of two wavelengths from the metasurface, an identical metasurface to that having been designed for sound focusing may be employed to convert a cylindrical wave to a plane wave. Figure 6a demonstrates the simulated resultant pressure profile in which the cylindrical wavefront originating from the monopole source has been reformed to yield a laterally confined plane waveform on the transmission side of the metasurface. In order to visualize the experimental wave pattern, the resultant pressures on the transmission side of the metasurface (shown in Fig. 6a) have been measured both with and without the presence of the metasurface (results shown in Fig. 6b, c). In Fig. 6b, the normalized pressure in the absence of the metasurface is shown in which the acoustic wavefront represents a diverging cylindrical waveform. Please note that although the experimental domain is bounded with absorbing foam to mitigate reflection, the impedance mismatch at the domain boundary resulted in minor and localized deviation of the pressure from the ideal form. Figure 6c demonstrates the resultant pressure field on the transmission side of the fabricated metasurface. The experimentally derived and normalized pressure field in the presence of the metasurface clearly demonstrates a laterally confined plane wavefront, which is in a good agreement with the pattern expected from the numerical solution, shown in Fig. 6a.

Acoustic wave focusing, cylindrical-to-plane wave conversion, and beam splitting represent simple examples of acoustic wavefront manipulation analyzed herein in order to demonstrate the capability of gradient space-coiling metamaterials. Beyond acoustic wavefront manipulation, including the realization of more complicated acoustic patterns, metasurfaces may also be employed to mitigate the effects of aberrant layers by aberration correction. In all these cases, the precise design of the metasurface is simplified by the capacity for modulating both phase and amplitude. Moreover, the added degree of freedom to modulate amplitude, in addition to phase, offers opportunities for performance that surpasses the capabilities of phase-based approaches (see Supplementary Note 4 for additional details).

## Discussion

The work presented herein, founded on the basis of a well-known acoustic metamaterial structure, namely space-coiling metamaterials, introduces a class of horn-like space-coiling metamaterials, which provides sufficient degrees of freedom for full acoustic wave control. Initially, the limitations of conventional space-coiling metamaterials for simultaneous phase and amplitude modulation are investigated, demonstrating that transmission through conventional space-coiling structures possesses topological-like bound, which is not frequency or unit cell dimension dependent. Moreover, we have demonstrated that this bound applies to any passive, lossless metasurface with real acoustic impedance and highlighted the importance of the reactance term for full wavefront modulation. Next, horn-like space-coiling structure capable of phase-amplitude modulation beyond conventional space-coiling structure limits is proposed and analyzed. Finally, metasurfaces featuring the proposed structures have been designed and simulated with the aims of sound focusing and acoustic beam splitting, whereas cylindrical to plane wave conversion has been experimentally validated. Horn-like space-coiling metamaterials offer a new methodology in metasurface design, in which phase and amplitude of the transmitted wave can simultaneously be modulated and tuned, yielding the capacity for complete wavefront shaping for myriad applications.

## Methods

**Experimental setup and procedure**. The acoustic metasurface with the dimensions detailed in Supplementary Table 1 (Focusing Metasurface) was fabricated with a commercial 3D printer (Dimension SST 1200es) from Acrylonitrile-Butadiene-Styrene plastic with a resolution of 0.2 mm. The experimental setup is shown in Fig. 7 and is composed of two thick plywood sheets of dimensions 250 cm × 250 cm × 2.5 cm placed in parallel with a spacing of 2 cm to create a 2D domain. Domain boundaries were confined with an absorbing foam 5 cm in thickness to mitigate the back reflection with an optimal performance at 1 KHz. In order to realize an acoustic point source within this domain, a loudspeaker was coupled to a narrow tube outlet to generate a localized point source. In order to ultimately map the acoustic field, a measurement region defined with appropriate offset from boundaries (~ 40 cm) was meshed with 190 equally spaced probing points with center-to-center spacing of 9 cm. The experimental procedure was designed to be performed at a frequency of 1 KHz and associated dimensions and spacing have been realized accordingly. To map the field within the measurement region, two initially calibrated microphones (Audix TM1) have been used in which one microphone is fixed throughout the entire mapping procedure and the second microphone moved to sample all probing points. At each probing location, the complex transfer function between the two microphone signals was calculated and averaged over 10 readouts (experimental data for both cases have been detailed in Supplementary Table 2). Finally, spline interpolation was utilized to visualize the resultant acoustic field in finer mesh.

**Numerical simulations**. All simulations were performed with finite element solver COMSOL Multiphysics using the pressure-acoustic module in the frequency-domain. In the lossy model simulations, the narrow region acoustic module has been used, which incorporates the effect of thermal and viscous losses. For obtaining the resulting transmission coefficient from each horn-like unit cell, the 2D wave-guide model, similar to the impedance tube, has been utilized and the transmission coefficients have been calculated using the microphone transfer function method[27]. The waveguide and all unit cells are considered perfectly rigid medium and the perfectly matched layer has been implemented to enclose the computational domain to mitigate the subsequent reflections.

**Data availability**. The authors declare that all data supporting the results of the study are available within the published article and its Supplementary Information section.

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

## Acknowledgements

We thank Boston University Materials Innovation Grant and Dean's Catalyst Award. We also thank the Boston University Photonics Center for technical support.

## Author contributions

R.G., S.A., and X.Z. conceived the idea and interpreted the results. R.G. designed and performed the theoretical and numerical calculations. J.N. helped with the numerical simulations. R.G. and J.N. under guidance of R.G.H. designed and performed the experiment. R.G., S.A. and X.Z. contributed to the preparation and writing of the manuscript, and X.Z. planned, coordinated, and supervised the project.

## Additional information

**Competing interests:** The authors declare no competing interests.

