## [Peer Review File · Nature Communications]

Reviewers' comments:

Reviewer #1 (Remarks to the Author):

The authors have proposed a new methodology by using Horn-like modification to space-coiling structures to construct acoustic metasurfaces. It allows transmission control on both amplitude and phase modulation based on a gradual change of channel width. Convincing numerical results, about focusing and collimation, are present to support the claim of their designs as well.

I hesitate on recommending its publication based on two reasons. First, based on that there are already works on acoustic metasurfaces, albeit mainly on phase control, the current work does not have significant advance over previous ones. For example, Applied Physics Letters 110, 191901 is about holograms using acoustic metasurfaces with independent decoupled control of both amplitude and phase, with experiment. Second, I think having amplitude control in addition to phase control is just a matter of design favour. If we have additional absorption, one can imagine the amplitude can be tuned in addition to phase, without insisting the lossless criteria imposed by the current work. However, I would say that the work is interesting on its own, and the model it provides is expected to be very useful for researchers in the field of acoustic metasurfaces.

Reviewer #2 (Remarks to the Author):

This article presents a unique and general contribution to the field of acoustic metamaterials, specifically subwavelength surfaces/interfaces known as acoustic metasurfaces that allow for the control of reflected or transmitted fields. Specifically, this work presents an analysis of the limits of standard coiled-up-space metasurfaces demonstrating that the control of the amplitude and phase of the fields transmitted through previously-designed metasurfaces are bounded to lie in a small region of the available magnitude and phase space of the complex-valued transmission coefficient. This restriction is shown to be due to the fact that previous coiled-space metasurface designs are limited to the case where the geometry of a given structure is essentially a repeating unit in the thickness of the metasurface. The authors show that by varying the geometry in the thickness direction of the space-coiling structure, that it is possible to access large portions of the magnitude-phase space of the transmission coefficient. The authors advance several analytical relationships that allow for the prediction of the magnitude and phase of the acoustic field transmitted through the generalized metasurface and check the results with finite element (FE) methods. Excellent agreement is shown between the FE results and the analytical bounds predicted by their method. The analytical relationships are very valuable since it provides the engineer/designer with a design tool to design the structure of a coiled-space metasurface without having to resort to numerous numerical simulations as has been done in previous studies. Further, the insight provided by this work has greatly generalized the concept of a coiled-up-space acoustic metasurface and it therefore has high value to the scientific community.

I therefore recommend this submission for publication in the journal with only one minor modification, which is detailed below.

Remark: In the second paragraph of the section entitled "Metasurfaces for wavefront manipulation with simultaneous ...," the authors seem to have used numerical techniques to design the coiled-up-space structures for sound focusing and cylinder-to-plane wave lenses. Why were the analytical techniques for creating metasurfaces to modify the magnitude and phase of the transmitted signal not employed? Focusing has several well-known phase profiles, such as the hyperbolic or quadratic phase profile. Given the generality of the expressions advanced in this work, it seems odd that they were not employed for the examples chosen. It's possible that the relationships were indeed employed and the

FE models were shown simply for validation. If that is the case, it was not clearly stated. Regardless, in a revision the authors need to clarify how the design of the coiled-space metasurface was arrived at and they should make use of the analytical expressions they derived in the other sections of the article.

We would like to thank the reviewers for their generous suggestions regarding our manuscript entitled “Horn-like space-coiling metamaterials: toward simultaneous phase and amplitude modulation”. We found them constructive, with their incorporation having led to an overall strengthening of the work. A detailed explanation and summary of the revisions that were undertaken are outlined below:

Reviewer #1:

“I hesitate on recommending its publication based on two reasons. First, based on that there are already works on acoustic metasurfaces, albeit mainly on phase control, the current work does not have significant advance over previous ones. For example, Applied Physics Letters 110, 191901 is about holograms using acoustic metasurfaces with independent decoupled control of both amplitude and phase, with experiment.”

We thank you for your insightful comments and thoughtful review. The referenced work presented in Applied Physics Letters 110, 191901 is based on the decomposition of the metasurface into two units, one responsible for tuning the phase and the other serving as an amplitude modulator. We thank you for highlighting this work and allowing us to draw some important comparisons. Notably, the structure presented in this Applied Physics Letters work exhibits a fundamental limitation that may significantly limit the metasurface performance. As it is demonstrated in the supplementary section of their work, in order to cover the entire range of phase and amplitude, the length of the CUC unit and PP unit must be around 0.5λ and 0.4λ , respectively. This requirement bounds the unit cell's width to sizes larger than $\lambda/2$, which can significantly influence the conversion outcome. In the full wave modulation (phase-amplitude), since the desired complex transmission is essentially obtained by discretizing the conjugated wave profile along the metasurface, the unit cell's width plays a critical role in conversion efficiency. In contradistinction, the horn-like space coiling structure that we present herein does not have such a limitation and the metasurfaces designed in our work were readily implemented with unit cells of width $\lambda/6$. In order to present a detailed description of the unit cell's width effect on the conversion efficiency, we have added section 4.1 in the Supplementary file of the revised manuscript that quantitatively demonstrates the effects of unit cell width on focusing performance.

Figure S7 readily demonstrates that for the case of focusing presented herein, power confinement of approximately 6.5 may be achieved with the unit cell's width of $\lambda/6$, compared to an inferior performance of 4.5 when the unit cell's width is $\lambda/2$. Overall, we believe that the horn-like space coiling structure presented herein, along with the presented analytical formulations, provides a strong, novel platform for successful, precise and efficient wavefront modulation.

“Second, I think having amplitude control in addition to phase control is just a matter of design favor. If we have additional absorption, one can imagine the amplitude can be tuned in addition to phase, without insisting the lossless criteria imposed by the current work. However, I would say that the work is interesting on its own, and the model it provides is expected to be very useful for researchers in the field of acoustic metasurfaces.”

We are in agreement that phase modulation is a powerful engineering tool in antenna theory and wavefront modulation. However, the capability of simultaneous phase and amplitude modulation offers a new degree of freedom, which results in gaining full control of the transmitted wave. While this added degree of freedom may seem trivial in straightforward cases such as focusing, it is found to lead to a significant improvement in more complex situations. In order to more clearly demonstrate the difference in performance between phase engineering and phase-amplitude modulation, we have added section 4.2 in the Supplementary file of the revised manuscript. This additional example compares the ideal outcome of these two approaches for the case of splitting an acoustic wave into two directions ($\pi/12$ and $-\pi/12$). In this example, the small angle between the two split beams ($\pi/6$) has rendered the phase-based modulation relatively ineffective, as the pressure between the split beams is not mitigated sufficiently. However, with phase-amplitude modulation, near ideal splitting may be realized in this case. For completeness of our work,

the metasurface has been also designed with the aim of acoustic beam splitting and has been added into the main body of the manuscript in the section entitled “Metasurfaces for wavefront manipulation with simultaneous phase and amplitude modulation”.

Finally, in the simulations with regards to sound focusing, cylindrical to plane wave conversion and the newly added acoustic beam splitting, we have added the results associated with lossy cases in Figures 5.d, 5.e and 5.f. The lossy results demonstrate that, due to the presence of thermal and viscous losses, the pressure amplitude is decreased and conversion efficiency is lowered, however, the general design approach remains valid.

We are happy that you found our model useful and would like to sincerely thank you for your time and constructive comments.

Reviewer #2:

“Remark: In the second paragraph of the section entitled "Metasurfaces for wavefront manipulation with simultaneous ...," the authors seem to have used numerical techniques to design the coiled-up-space structures for sound focusing and cylinder-to-plane wave lenses. Why were the analytical techniques for creating metasurfaces to modify the magnitude and phase of the transmitted signal not employed? Focusing has several well-known phase profiles, such as the hyperbolic or quadratic phase profile. Given the generality of the expressions advanced in this work, it seems odd that they were not employed for the examples chosen. It's possible that the relationships were indeed employed and the FE models were shown simply for validation. If that is the case, it was not clearly stated. Regardless, in a revision the authors need to clarify how the design of the coiled-space metasurface was arrived at and they should make use of the analytical expressions they derived in the other sections of the article.”

Indeed, the structure of the horn-like space coiling metamaterial has been designed analytically. In order to design the metasurface targeting any particular functionality, initially, the desired complex transmission at each unit cell was calculated. In this step, we employed numerical methods since it can be readily generalized to any complicated case, such as a 2D hologram. The drawback of using the well-known phase profile for focusing is the lack of information regarding the optimized amplitude. These profiles have been demonstrated to serve best when phase modulation is the only degree of freedom. In our next step, having derived the required complex transmission at each unit cell, the appropriate internal structure was designed analytically. For this step, we took advantage of the availability of the analytical expression and, by sweeping through possible unit cell geometries, we generated the large database of structures and their associated complex transmission that served as our metasurface design library. Finally, after determining the internal structure of the unit cells analytically, a finite element model was used to validate

and visualize the structure's performance. The table at the end of supplementary section essentially summarizes each unit cell's structure and its associated analytical and numerical complex transmission.

For clarity, portions of the text in the section of the main body of the manuscript entitled "Metasurfaces for wavefront manipulation with simultaneous phase and amplitude modulation" have been revised to specify this design process and the manner by which the analytical approach has been utilized for our metasurface design.

We would like to sincerely thank you for your time and your constructive comments.

Reviewers' comments:

Reviewer #1 (Remarks to the Author):

I am still not convinced that the work is making a significant impact. For the issue of the independent control of amplitude and phase in the Applied Physics Letters 110, 191901, the current authors raised the disadvantage of the realized sample that the size is not small enough. I do not see it is very difficult to downsize it to like $\lambda/4$. It is using a very similar design as the current authors and more importantly, they have realized this independent control of amplitude and phase experimentally. The next step should lie in the experimental demonstration of applications of this independent control, for example, a high quality hologram with fine resolution, rather than a numerical study to relax one geometric parameter (an inhomogeneous channel width).

For the other issue about methodology, I agree with the authors that their design has their own merit. However, I still think that there can be different methods to achieve control of both amplitude and phase. The current approach is just too specific: we do not necessarily need the approach shown in this work and I do not see the generality of the current theory to benefit other studies. However, I admit that the question of the design is only a matter of taste. The model is very nice and I leave to the editor to decide whether this is enough for publication.

Reviewer #2 (Remarks to the Author):

The authors have responded to my concerns with the original manuscript. I recommend the current manuscript for publication in its current form.

We would like to thank the reviewers for their generous suggestions regarding our manuscript entitled “Horn-like space-coiling metamaterials: toward simultaneous phase and amplitude modulation”. We found them constructive, with their incorporation having led to an overall strengthening of the work. A detailed explanation and summary of the revisions that were undertaken are outlined below:

Reviewer#1:

"I am still not convinced that the work is making a significant impact. For the issue of the independent control of amplitude and phase in the Applied Physics Letters 110, 191901, the current authors raised the disadvantage of the realized sample that the size is not small enough. I do not see it is very difficult to downsize it to like $\lambda/4$. It is using a very similar design as the current authors and more importantly, they have realized this independent control of amplitude and phase experimentally. The next step should lie in the experimental demonstration of applications of this independent control, for example, a high quality hologram with fine resolution, rather than a numerical study to relax one geometric parameter (an inhomogeneous channel width)."

We thank you for your insightful comments and thoughtful review. Applied Physics Letters 110, 191901 reports a composite metasurface structure that can individually modulate phase and amplitude of the transmitted wave using CUC (phase modulator) and PP (amplitude modulator) units. The presented structure in this prior work is essentially a waveguide with four different sections in which several constraint with regards to the length and the cross-sectional areas of each part have been imposed to decouple the phase and amplitude of the transmission coefficient. The main drawback of their study is that the presented design methodology significantly relies on finite element simulation. Moreover, scaling down the unit cell size, although possible, necessitates an intensive trial and error FEM procedure in order to determine the proper coupler structure, since it needs to be coiled for widths smaller than $\lambda/2$ and, consequently, correction factors need to be recalculated throughout the equations. We believe that our manuscript and our design methodology differs from the referenced prior work in two major aspects. Firstly, a full analytical study using both TMM method and an equivalent model have been presented in our work in order to provide a powerful and simple design tool, without the need for the finite element method. Second, the conclusion of our work is more general than the Applied Physics Letters 110, 191901 work. In our manuscript, it is demonstrated that for full wavefront modulation, acoustic reactance is a key element. We have explained that one possible way to achieve complex acoustic impedance is a change in cross-section, which inspired us to modify the conventional space coiling structure accordingly and realize full wavefront modulation. The APL paper is focused on a single specific design that is encompassed by the more general theoretical analysis which we present and that has not been reported to date. Please note that complex acoustic impedance may also be realized with other methods such as internal resonance and, based on the conclusions of our manuscript, can also be utilized for full wave manipulation.

In the revised manuscript, we have performed experimental validation for the reverse case of focusing in which a cylindrical wavefront from a point source has been shaped to yield a laterally confined plane wave. The experimental results with regards to the wave pattern for both cases of with and without the presence of the metasurface have been added in the results section (Figure 6) and comprehensive details of the experimental procedure are added in the method section.

“For the other issue about methodology, I agree with the authors that their design has their own merit. However, I still think that there can be different methods to achieve control of both amplitude and phase. The current approach is just too specific: we do not necessarily need the approach shown in this work and I do not see the generality of the current theory to benefit other studies. However, I admit that the question of the design is only a matter of taste. The model is very nice and I leave to the editor to decide whether this is enough for publication.”

In our manuscript, we have presented a very general bound on the complex transmission that can be applied to any metasurface with any internal structure. We have demonstrated that transmission through any metasurface with real effective acoustic impedance essentially lays within a small complex region (governed by Equation 13) and acoustic reactance is a key element to realizing full wavefront modulation. Effective acoustic reactance can be realized with different approaches including the gradual change in the wave guide cross section, which is considered in our manuscript. In our work, we have used a space coiling metamaterial structure as an example to demonstrate this bound and, given the presented general theory in our work, we have modified this structure and demonstrate that full wavefront control may be realized. The focus of this work is to explore the theory and derive the governing analytical relations, with a numerical method employed solely to validate the theoretical results. The major finding of this paper is the complex bound discussed in our work, which has not been reported to date, and which we believe will be of great interest to the general research community. We would like to sincerely thank you for your time and consideration.

Reviewer #2:

“The authors have responded to my concerns with the original manuscript. I recommend the current manuscript for publication in its current form.”

We would like to thank you for your generous suggestion. We believe that your insightful review and thoughtful comments has significantly improved our work.

REVIEWERS' COMMENTS:

Reviewer #1 (Remarks to the Author):

In light of the new experimental results to support the theory claim, I am glad to see its publication .

Reviewer 1.

“In light of the new experimental results to support the theory claim, I am glad to see its publication.”

Author response:

We would like to thank you for all your generous suggestions. We believe that your insightful review and thoughtful comments has significantly improved our work.